# Re-Examining Summarization Evaluation across Multiple Quality Criteria

**Ori Ernst**[* 1,2]**, Ori Shapira**[1]**, Ido Dagan**[2]**, and Ran Levy**[1]

[1]Amazon        [2]Bar-Ilan University
oriern@gmail.com
{orishap, ranlevy}@amazon.com
dagan@cs.biu.ac.il

## Abstract

The common practice for assessing automatic evaluation metrics is to measure the correlation between their induced system rankings and those obtained by reliable human evaluation, where a higher correlation indicates a better metric. Yet, an intricate setting arises when an NLP task is evaluated by *multiple* Quality Criteria (QCs), like for text summarization where prominent criteria include relevance, consistency, fluency and coherence. In this paper, we challenge the soundness of this methodology when multiple QCs are involved, concretely for the summarization case. First, we show that the allegedly best metrics for certain QCs actually do not perform well, failing to detect even drastic summary corruptions with respect to the considered QC. To explain this, we show that some of the high correlations obtained in the multi-QC setup are spurious. Finally, we propose a procedure that may help detect this effect. Overall, our findings highlight the need for further investigating metric evaluation methodologies for the multiple-QC case.

## 1 Introduction

The broad interest in text generation has triggered significant progress in the development of respective automatic evaluation metrics, complementing costly manual evaluation. In order to assess and rank various automatic metrics, the conventional setup compares the ranking of various systems according to metric scores and human scores, where a high correlation between them indicates an effective metric. In some cases, a task requires evaluation by multiple quality criteria (QCs), as is prominently the case for text summarization, where common criteria include coherence, fluency, faithfulness and saliency. In such cases, comparing metric scores to human scores separately for each criterion should supposedly determine which metrics

---

* Work was done as an intern at Amazon.

| Metric | Coherence | Consistency | Fluency | Relevance |
|---|---|---|---|---|
| ROUGE-3 | 0.2206 | **0.7059** | 0.5092 | 0.3529 |
| CHRF | **0.3971** | 0.5294 | 0.4649 | **0.5882** |
| METEOR | 0.2353 | 0.6324 | **0.6126** | 0.4265 |

Table 1: Correlations (Kendall's $\tau$) between metric and human scores for each QC, as taken from Fabbri et al. (2021). Only the best metric in each QC is presented.

are most suitable for evaluating each respective criterion.

To enable such a metric evaluation procedure, some annotation efforts have been conducted (e.g., Bhandari et al., 2020; Fabbri et al., 2021), where system summaries were manually scored according to several QCs. This yielded a "gold ranking" of systems per QC, to which metric rankings can be compared. The recent SummEval benchmark (Fabbri et al., 2021) has attracted interest due to its thorough data collection of 1,700 annotated system summaries over 17 systems. Overall, all summaries were rated over 4 different QCs: Relevance, Consistency, Fluency and Coherence (clarified in §2). Within this benchmark, the majority of the evaluated metrics were primarily intended to assess Relevance, with a smaller subset comprising statistical analysis-based metrics such as summary length, which were not specifically designed to measure any of the four aforementioned QCs. In our work, we focus on the major group of metrics that were designed to measure Relevance. Table 1 presents correlations between the best evaluation metrics in each QC and the SummEval human scores. The full table can be found in Table 6 in Appendix A.1.

Surprisingly, a closer examination of metric performances in SummEval reveals that many of the high correlations are unlikely to imply high fitness of the best performing metrics and the respective QC. For example, ROUGE-1 (Lin, 2004), which measures the lexical match of unigrams between the system and reference summaries, appears to

perform well as a fluency metric. Fluency is typically not determined based on unigrams alone and should not depend on the reference summary. This raises questions such as (1) what was the cause for these high correlations, and (2) when can correlations be trusted.

In this paper we address these two questions with thorough analyses providing two contributions. First, we point out that the conventional scheme for metric ranking is problematic in the case of multiple quality criteria. To achieve this we discovered that even the best metrics for each QC fail to penalize system summaries containing corruptions that should adversely affect that QC. This suggests that these metrics are ineffective in measuring the intended QC. In addition, we show that there are *spurious* correlations in a multi-QC setup like summEval, caused by high performance correlations across criteria. Second, we suggest a method for detecting metric-to-human correlation scores that are suspected as spurious, by removing the effect of a *confounding variable*, which reveals a performance degradation in many cases. This provides a first step to cope with this obstacle, while calling for further research on metric evaluation in the multi-QC case.

## 2 Background and Related Work

A high-quality summary should satisfy several requirements, including preserving the salient information of the source, being faithful to the source, and being coherent. The DUC benchmarks (NIST, 2002) made an early attempt at capturing these requirements with a human evaluation procedure that assessed several readability qualities as well as content responsiveness (Dang, 2005). Later, additional benchmarks focused on content quality (e.g., Bhandari et al. (2020), TAC (NIST, 2008)) or linguistic quality (Chaganty et al., 2018) separately. Kryscinski et al. (2019) laid out four Quality Criteria (QCs) on which to assess summary quality, which were reinforced in the SummEval summary evaluation benchmark (Fabbri et al., 2021), annotated over the CNN/DailyMail (Nallapati et al., 2016) summarization dataset.

Described briefly, the four QCs are: **Relevance**, measuring the importance of summary content with respect to the source; **Consistency**, measuring the faithfulness of the summary content to the source; **Fluency**, measuring the linguistic quality of individual sentences in the summary; and **Coherence**, measuring the quality of the collective structure of the sentences in the summary. A summarization system is rated for each QC by averaging human scores (1 to 5 scale) over all input instances. Thus, multiple systems can be scored and ranked separately for each QC in accordance with the respective human scores.

To assess the effectiveness of an automatic evaluation metric, it is first applied on the outputs of several summarization systems, yielding a respective score per system (averaged over all instances). The metric performance for a certain QC is then measured as the correlation between these metric scores and the corresponding human scores for that QC. The described meta-evaluation procedure enables to rank multiple evaluation metrics per each QC separately, as illustrated in Table 1. In our work, we question whether highly ranked metrics on a particular QC are indeed suitable for evaluating that QC, claiming that such correlation to human scores are not necessarily indicative in the multi-QC setting.

Some studies examined other aspects of this meta-evaluation scheme. Peyrard (2019) and Bhandari et al. (2020) showed that meta-evaluation using old systems or datasets, like those of DUC and TAC, yield erroneous trends on modern summarization benchmarks. Deutsch et al. (2021) investigated the preciseness of correlations between metrics and human annotations in meta-evaluation benchmarks, and proposed approaches to improve the level of confidence. Finally, Deutsch et al. (2022) discussed ways of improving the reliability of system-level correlations in meta-evaluation.

## 3 Metric Robustness to Summary Corruptions

In this section, we aim to investigate whether a metric that was found to strongly correlate with human scores for a certain QC according to SummEval reliably measures this specific QC. Specifically, we suggest that the performance of metrics – that were designed to measure Relevance – on other QCs (e.g., using ROUGE to measure Fluency) may be questionable. To examine this, we artificially corrupt SummEval (Fabbri et al., 2021) system summaries in various forms, each specifically designed to degrade a single QC. The corruption is expected to have a large impact on the corresponding QC measurements, while having minimal effect on other QCs. Accordingly, we may conclude that

a metric that fails to penalize corrupted summaries does not actually measure the *specific* QC for which corruption was introduced.

In what follows, we experiment with the following corruptions of the system summaries with respect to each QC (except for Relevance which is the QC most metrics were designed to capture). **Fluency:** All verbs are replaced with their lemma form, resulting in ungrammatical sentences. **Coherence:** Sentences are randomly shuffled to disrupt the structure of the summary. This corruption is inspired by the Shuffle Test (Barzilay and Lapata, 2008) used to evaluate whether models can detect incoherent text. **Consistency:** All PERSON named entities (using SpaCy NER, Honnibal et al., 2020) are replaced with different PERSONs from the source document. This is in fact a common factual mistake of models (Pagnoni et al., 2021). An example for each corruption type can be found in Table 9 in the Appendix.

To test the meta-evaluation of metrics presented in SummEval, we examine the sensitivity of the best metric for each QC, according to SummEval results (from Table 1), to the QC-specific corruption. More specifically, for each QC, we ranked all systems with the best metric, corrupted each system in turn, and examined whether that system's ranking was subsequently downgraded below other systems that were initially scored significantly lower. We found that none of the corrupted systems were downgraded in such relative rankings with the Coherence and Fluency corruptions, while only two (out of 17) were downgraded with the Consistency corruption. Assuming our drastic corruptions should have caused a ranking downgrade, these results indicate that the top performing metrics for Coherence, Fluency and Consistency were mostly unable to penalize the corrupted system summaries, suggesting they are not sufficiently reliable at measuring these QCs.

To validate the aforementioned assumption, that our corruptions should actually cause a ranking downgrade, we manually annotated a sample of 9 systems that contain corrupted summaries and found that all system rankings were downgraded after corruption. (More details can be found in Appendix B.) Since the best automatic metric did not reflect this change in ranking, we conjecture that SummEval meta-evaluation scores, even when they appear to be high, are not reliable. In the next section, we investigate the possibility that this

| Anchor
QC | Coherence | Consistency | Fluency | Relevance |
|---|---|---|---|---|
| Coherence | 1.00 | .28 / .24 / .11 | .48 / .25 / .06 | .69 / .53 / .27 |
| Consistency | .28 / .24 / .07 | 1.00 | .60 / .36 / .11 | .41 / .25 / .07 |
| Fluency | .48 / .25 / .08 | .60 / .36 / .18 | 1.00 | .61 / .21 / .09 |
| Relevance | .69 / .53 / .21 | .41 / .25 / .06 | .61 / .21 / .04 | 1.00 |

Table 2: Kendall's $\tau$ correlations between human QC ratings on the system-level/instance-level/bucketing. The columns serve as anchors solely for bucketing. Applying bucketing diminishes correlation between QCs.

is caused by *spurious correlations*, and propose a possible means to identify them.

# 4 Analysis of Spurious Correlations

A possible explanation for the contradicting findings that "high performing" metrics fail to penalize corrupted summaries, is the high correlations observed between human scores of different QCs (termed $correlation_{human}$), as seen in system-level scores in Table 2 (left-most figure in each cell). As a result, high correlations between metrics and human scores on all QCs (termed $correlation_{metric}$) are usually due to the high $correlation_{metric}$ with one confounding QC combined with the strong $correlation_{human}$ between this confounding QC and the remaining QCs. As all of the aforementioned best metrics were initially designed to measure Relevance, we conjecture that the Relevance QC acts as a main *confounding variable*, making other $correlations_{metric}$ *spurious*. Although spurious $correlations_{metric}$ can be found in different setups, they are more likely in multi-QC setups, like summarization. In such a setup, models are typically optimized to excel in all QCs, resulting in enhanced systems across all QCs that consequently yield high $correlations_{human}$ among these QCs.

Next, we suggest a systematic method to detect a confounding QC that undermines $correlations_{metric}$ with other QCs. To that end, we propose to remove the effect of each QC, in turn, on the remaining QCs by annulling the $correlations_{human}$ between them. To do this, we calculated $correlations_{metric}$ over small subsets (i.e., buckets) of instances in which the annulled QC has low variance. We show that for most metrics when the $correlation_{human}$ to Relevance is annulled, the $correlations_{metric}$ to Fluency, Coherence and Consistency drops drastically, while the $correlations_{metric}$ to Relevance is immune to similar manipulations of the other three QCs. This suggests that Relevance is indeed a confounding factor for the other QCs.

To compute $correlations_{metric}$ within buck-

ets we first note that the original SummEval correlations$_{metric}$ were calculated at the system level. That is, the 100 instance scores per system are averaged to produce 17 data points for correlation. However, dividing these 17 data points into buckets results in statistically unreliable correlation scores inside each bucket due to the limited number of data points in it. To address this, we utilized the instance-level correlation approach (Bojar et al., 2017; Freitag et al., 2021), which incorporates all 1,700 instance scores as individual data points without averaging per-system. Dividing the entire 1,700 instances into buckets ensures a sufficient number of data points within each bucket. Since confounding factors are inherent to the data, identifying a confounding variable at the instance-level implies its presence at the system level as well.

Inspired by *stratification* analysis (Mantel and Haenszel, 1959), in order to remove the effect of a particular QC (termed *anchor* QC) to assess its potential to be a confounding factor, we divide the system summary instances, which are associated with human score tuples of <Relevance, Consistency, Fluency, Coherence>, into *buckets*. Each bucket contains tuples with roughly equal scores of the anchor QC. Since scores are on a 1-to-5 scale, we use 5 buckets. As an example, if we would like to anchor Relevance, the first bucket will contain tuples with Relevance $\approx 1$. Accordingly, the correlation$_{human}$ inside each bucket between the anchor QC human scores and each other QC degrades substantially. As Table 2 shows, averaging these low correlations$_{human}$ over all 5 buckets and weighting by bucket size, results in "bucketing" value in each cell that reduces the initial instance-level correlations$_{human}$ between QCs by 2-5 times.

Next, we used this approach to calculate the correlations$_{metric}$ inside each bucket, thus neutralizing the effect of the anchor QC. Again, the five bucket correlations$_{metric}$ are averaged and weighted by bucket size. Finally, to measure whether the obtained bucketing value has changed significantly with respect to the original instance-level correlation$_{metric}$, we calculate the absolute relative difference between the two scores. A high relative difference means that the correlation$_{metric}$ has changed significantly after removing the anchor QC. This undermines the reliability of the original correlation$_{metric}$ scores, and suggests the anchor QC as a confounding factor. While our work does not provide a specific threshold for detecting spurious correlation$_{metric}$ based on relative difference, this process does alert for potential unreliability when the relative difference is relatively high.

The relative difference scores by each one of the anchor QCs and each metric are shown in Appendix A.3. To summarize these detailed findings, we focused on the majority of metrics that were designed to measure Relevance. For each anchor and evaluated QC, we computed the median relative difference of all metrics. As can be seen in Table 3, we observe that the largest relative differences occur when Relevance serves as the anchor QC, as presented in the blue column. This means that the original correlations$_{metric}$ to Coherence, Fluency and Consistency were strongly affected by Relevance, as a strong confounding variable. However, when Relevance serves as the evaluated QC, as demonstrated in the yellow row, the relative differences are quite low, regardless of the anchor QC. This means that other QCs are probably not confounding variables for Relevance.

We also used the bucketing analysis to evaluate two other metric groups that roughly estimate other QCs, and observed the same phenomenon. The first group contains metrics that measure the percentage of repeated n-grams in the system summary. As a summary with repeated information is less coherent, these metrics are more suitable as rough estimates for Coherence. Accordingly, Table 4 shows high relative differences when Coherence functions as the anchor (marked as a blue column), meaning that when neutralizing Coherence, the correlations$_{metric}$ to other QCs change dramatically. On the other hand, when other QCs function as anchors, the correlation$_{metric}$ with Coherence is almost unchanged. This is expressed by the low relative difference (marked as the yellow row). Overall, this analysis suggests that Coherence is the confounding factor of this metric group and the original correlations$_{metric}$ are spurious.

The second group contains metrics measuring the percentage of novel n-grams in the summary that are not found in the input document. These metrics capture abstractiveness, making them potentially useful as rough (negative) estimates for Consistency and Fluency. This is due to the capability of these metrics to identify extractive summaries, which inherently possess consistency and fluency. Accordingly, we show the same phenomenon in Table 5 where bucketing by Consistency or Fluency as anchors yields high relative

| QC \ Anchor | Coherence | Consistency | Fluency | Relevance |
|---|---|---|---|---|
| Coherence | N/A | .20 (.17-.23) | .20 (.14-.24) | .82 (.74-.94) |
| Consistency | .25 (.20-.34) | N/A | .30 (.18-.49) | .66 (.53-.70) |
| Fluency | .50 (.40-.59) | .62 (.43-.72) | N/A | .72 (.44-.85) |
| Relevance | .18 (.13-.24) | .15 (.11-.20) | .08 (.03-.16) | N/A |

Table 3: Median (quantile 25-quantile 75) of the absolute relative difference between original correlation and bucketing correlation, over metrics that were designed to measure Relevance.

| QC \ Anchor | Coherence | Consistency | Fluency | Relevance |
|---|---|---|---|---|
| Coherence | N/A | .14 (.09-.14) | .18 (.16-.25) | .14 (.07-.14) |
| Consistency | .79 (.74-.89) | N/A | .58 (.51-.68) | .38 (.35-.44) |
| Fluency | .73 (.71-.78) | .46 (.45-.49) | N/A | .15 (.14-.23) |
| Relevance | .76 (.72-1.06) | .31 (.23-.58) | .37 (.33-.48) | N/A |

Table 4: Median (quantile 25-quantile 75) of the absolute relative difference between original correlation and bucketing correlation, over metrics that roughly measure Coherence.

| QC \ Anchor | Coherence | Consistency | Fluency | Relevance |
|---|---|---|---|---|
| Coherence | N/A | .78 (.75-1.05) | .82 (.78-0.87) | .32 (.26-.44) |
| Consistency | .05 (.03-.09) | N/A | .22 (.22-.25) | .11 (.07-.12) |
| Fluency | .04 (.02-.13) | .39 (.38-.45) | N/A | .07 (.06-.09) |
| Relevance | .52 (.40-.76) | .79 (.55-2.52) | .96 (.85-1.88) | N/A |

Table 5: Median (quantile 25-quantile 75) of the absolute relative difference between original correlation and bucketing correlation, over metrics that roughly measure Consistency and Fluency.

differences (blue columns), while bucketing by other QCs leaves low relative differences to Consistency and Fluency (yellow rows). As in this case we found two confounding factors, we conjecture that there is another unmeasured human QC that assesses abstraciveness directly that eventually influences Consistency and Fluency. In such a case, this abstraciveness QC therefore functions as a confounding factor.

Overall, this analysis points out the problematic conventional scheme of metric evaluation in a multi-QC setting. We found that except for the QC that the metrics were designed to measure, most of the correlations_metric to other QCs are spurious. Further exploration of adjusting this analysis for other scenarios, such as cases involving two confounding factors, is left as future work.

It is worth noting that spurious correlations can alternatively be detected with the *partial correlation* approach (Whittaker, 2009). The confounding variable is neutralized by calculating the correlation between the residuals resulting from the linear regression of each of the variables to the confounding variable. The partial correlation scores in our setting indeed display the same trend as our bucketing method, detecting Relevance as the confounding variable in most metrics. In contrast to the partial correlation approach, bucketing has the advantages of being more interpretable and not assuming linear dependency between variables. See Appendix C for an analysis and comparative discussion between the methods.

## 5  Conclusion

We challenged the conventional manner of evaluating and ranking summarization metrics according to correlation with human scores in the multi-QC setting. We found that human ratings over recent state-of-the-art systems tend to correlate between different QCs, which leads to unreliable metric performance scores. To demonstrate this, we showed that metrics that allegedly measure a QC, negligibly decrease their score for corrupted summaries with respect to this QC. To cope with this obstacle, we proposed a bucketing method that removes the effect of the confounding variable and detects unreliable correlations.While this work mostly highlights the problem, we strongly encourage the development of additional approaches to tackle it.

## Acknowledgments

We would like to thank Rotem Dror, Amir Feder and the paper reviewers for their comments, and Alex Fabbri for his support in reconstructing the SummEval results. The work described herein was supported in part by the Katz Fellowship for Excellent PhD Candidates in Natural and Exact Sciences and by the Israel Science Foundation (grant no. 2827/21).

## Limitations

This study highlights a phenomenon that occurs when assessing summarization metrics across varying quality criteria. The findings are empirically shown only on SummEval, which is a relatively large-scale and high-quality meta-evaluation benchmark. Furthermore, there do not exist other major benchmarks that would enable a similar analysis. Nevertheless, the findings would be further strengthened if they could be examined on additional benchmarks.

Additionally, although our analysis offers strong empirical evidence that the Relevance QC is the confounding variable in most metrics in the Sum-

mEval setting, there could be other *external* factors that cause the strong correlations among the QCs.

We also rely, to a certain degree, on logical intuition and understanding of the proposed metrics in order to convince the reader of our findings. For example, it is very reasonable to assume that certain summarization metrics do not actually have the ability to measure a specific QC. In the case of ROUGE-1, there should not be a true relationship between the number of overlapping unigrams with another text and the Fluency of the evaluated text. Any corresponding chance correlation is presumably not due to a direct intent of the metric.

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

## A  Experiment Technicalities

### A.1  Running Systems and Metrics

All of our experiments were based on the resources provided by the SummEval benchmark (in the GitHub repository at `https://github.com/Yale-LILY/SummEval`). System summaries and human scores were present in the GitHub repository. We ran the metrics on summaries using the code provided in the repository, with a few minor adaptations. The original correlation table of SummEval is presented in Table 6.

### A.2  Corruptions

An example of all corruption types is presented in Table 9.

### A.3  Bucketing

To map the 1-5 scores into 5 buckets we rounded the scores to the nearest integer. All metrics' bucketing correlations for each anchor are presented in Tables 10, 11, 12, 13.

To compute the absolute relative difference between the original correlations$_{metric}$ and the bucketing correlations$_{metric}$, we first computed the absolute values of each correlation score. This allowed us to assess the metric's ability to capture the human scores, whether in a positive or negative relationship. Next, we calculated the absolute difference between the two correlation values. A high absolute difference indicates a significant modification in the original correlation after bucketing, either in an upward or downward direction, highlighting the unreliability and spurious nature of the original correlation. While the majority of cases showed a positive difference, indicating that the original correlation was higher than the bucketing correlation, there were rare instances where the difference was negative. A negative difference implies that the original correlation was initially low but experienced a significant increase after bucketing.

## B  Manual Annotation for Corrupted Systems

In Section 3, we aim to validate the assumption that substantially corrupted systems should be penalized in human ranking. To support this claim, we conducted a manual annotation process on a randomly selected subset of 20 documents from a total of 100. Specifically, for each QC, we chose three corrupted systems (with 20 documents) that were not identified for degradation by the best automatic

| Metric | Coherence | Consistency | Fluency | Relevance |
|---|---|---|---|---|
| ROUGE-1 | 0.2500 | 0.5294 | 0.5240 | 0.4118 |
| ROUGE-2 | 0.1618 | 0.5882 | 0.4797 | 0.2941 |
| ROUGE-3 | 0.2206 | **0.7059** | 0.5092 | 0.3529 |
| ROUGE-4 | 0.3088 | 0.5882 | 0.5535 | 0.4118 |
| ROUGE-L | 0.0735 | 0.1471 | 0.2583 | 0.2353 |
| ROUGE-su* | 0.1912 | 0.2941 | 0.4354 | 0.3235 |
| ROUGE-w | 0.0000 | 0.3971 | 0.3764 | 0.1618 |
| ROUGE-we-1 | 0.2647 | 0.4559 | 0.5092 | 0.4265 |
| ROUGE-we-2 | -0.0147 | 0.5000 | 0.3026 | 0.1176 |
| ROUGE-we-3 | 0.0294 | 0.3676 | 0.3026 | 0.1912 |
| $S^3$-pyr | -0.0294 | 0.5147 | 0.3173 | 0.1324 |
| $S^3$-resp | -0.0147 | 0.5000 | 0.3321 | 0.1471 |
| BertScore-p | 0.0588 | -0.1912 | 0.0074 | 0.1618 |
| BertScore-r | 0.1471 | 0.6618 | 0.4945 | 0.3088 |
| BertScore-f | 0.2059 | 0.0441 | 0.2435 | 0.4265 |
| MoverScore | 0.1912 | -0.0294 | 0.2583 | 0.2941 |
| SMS | 0.1618 | 0.5588 | 0.3616 | 0.2353 |
| BLEU | 0.1176 | 0.0735 | 0.3321 | 0.2206 |
| CHRF | **0.3971** | 0.5294 | 0.4649 | **0.5882** |
| CIDEr | 0.1176 | -0.1912 | -0.0221 | 0.1912 |
| METEOR | 0.2353 | 0.6324 | **0.6126** | 0.4265 |
| SummaQAˆ | 0.1176 | 0.6029 | 0.4059 | 0.2206 |
| BLANCˆ | 0.0735 | 0.5588 | 0.3616 | 0.2647 |
| SuPERTˆ | 0.1029 | 0.5882 | 0.4207 | 0.2353 |
| Novel unigramˆ | 0.1471 | -0.2206 | -0.1402 | 0.1029 |
| Novel bi-gramˆ | 0.0294 | -0.5441 | -0.3469 | -0.1029 |
| Novel tri-gramˆ | 0.0294 | -0.5735 | -0.3469 | -0.1324 |
| Repeated unigramˆ | -0.3824 | 0.1029 | -0.0664 | -0.3676 |
| Repeated bi-gramˆ | -0.3824 | -0.0147 | -0.2435 | **-0.4559** |
| Repeated tri-gramˆ | -0.2206 | 0.1471 | -0.0221 | -0.2647 |

Table 6: SummEval Kendall's $\tau$ correlations between metrics and human annotations for each QC (taken from (Fabbri et al., 2021)). ˆ denotes reference-free metrics. The most-correlated metric in each column is in bold.

metric as described in Section 3. These systems were then annotated against sampled lower-ranked systems, ensuring a ranking difference of at least 6 places (above one-third of the number of systems) based on the best automatic metric. Additionally, we confirmed that the corrupted systems, when not corrupted, achieved higher rankings compared to the lower-ranked systems according to summEval manual annotation. Finally, for each QC, we annotated 3 pairs of systems, consisting of one corrupted system (corrupted with the QC-specific corruption) and one lower-ranked uncorrupted system.

During the annotation process, we compared each system pair in terms of the relevant QC at the instance level, and aggregated the results to identify the system with more preferred instances. Specifically, for a specific pair of systems, the annotator gets two system summaries (one per system) of the same document, and should select the best summary in terms of the corrupted QC. After annotating all pairs, the system with the most preferred summaries is considered better than the other system.

For the annotation of Coherence, one of the au-

| QC
Sys Pair | Coherence | Consistency | Fluency |
|---|---|---|---|
| I | 65 | 60 | 100 |
| II | 80 | 50 | 100 |
| III | 75 | 75 | 100 |
| **Mean** | **73** | **62** | **100** |

Table 7: Percent of the uncorrupted system summaries that were manually preferred in each pair of systems. Notice that the system pairs are different for each QC, therefore the columns are not comparable.

| Anchor
QC | Coherence | Consistency | Fluency | Relevance |
|---|---|---|---|---|
| Coherence | N/A | .21 (.16-.24) | .11 (.09-.13) | .87 (.63-.90) |
| Consistency | .22 (.19-.28) | N/A | .14 (.10-.22) | .42 (.36-.46) |
| Fluency | .43 (.36-.50) | .63 (.40-.80) | N/A | .63 (.35-.86) |
| Relevance | .19 (.15-.30) | .11 (.10-.14) | .03 (.03-.09) | N/A |

Table 8: Median (quantile 25-quantile 75) of the absolute relative difference between original correlation and partial correlation, over metrics that were designed to measure Relevance.

thors annotated all 3 system pairs. However, since the Fluency and Consistency corruptions (lemmatizing all verbs and replacing all PERSON entities with others) can be easily noticeable by the authors of the paper, we used Amazon Mturk[1] workers for their annotation. We used workers from a list of 90 pre-selected workers from English speaking countries. These workers accomplished high quality work in other NLP-related tasks we have conducted in the past. Each pair of summaries was annotated by three workers, and the majority vote was considered for the final score.

Table 7 presents the rate of uncorrupted low-ranked system summaries that were preferred over the corrupted summaries in each system pair. As can be measured clearly, in all pairs the uncorrupted summaries were preferred in 50 percent or more although they were ranked lower prior to corruption. This indicates that the corrupted systems indeed should be downgraded in their ranking. The conclusion is therefore that our corruptions are effective in degrading the corrupted system's ranking.

## C  Partial Correlation

When examining the statistics literature for methods for detecting spurious correlations, we focused on the two prominent approaches: partial correlation (Whittaker, 2009) and data stratification (Mantel and Haenszel, 1959). Our bucketing method, presented in Section 4, is a form of stratification.

---

[1] https://www.mturk.com

Methodologically, we identified two advantages of the bucketing approach in comparison to partial correlation. First, bucketing arranges the data into sets (buckets), where the confounding variable is neutralized in each set. This allows for further interpretability and analysis of the data, e.g., examining correlations and other statistics within each set. Meanwhile the partial correlation method only provides a final score without any further interpretation of the score or of the data. In other words, while provides debiased correlation, the bucketing method provides debiased data. Second, partial correlation is based on linear regression between the assessed variables and the confounding variable, and hence assumes a potentially linear correlation between these variables. Bucketing, on the other hand, generally allows the neutralization of any type of spurious correlation, making this method more robust for future extensions (even if in the current work we measured only linear correlations within each bucket).

Empirically, the partial correlation scores show the same trend as the bucketing scores. Table 8 displays the median (with the 25th and 75th percentiles) differences between the original and partial correlations$_{metric}$ for a group of metrics designed to measure Relevance, using the same format as described in Section 4 for the bucketing method. Similar to the bucketing method, we observe significant differences in cases where Relevance serves as the anchor QC (the column marked in blue), while minimal differences are observed for the Relevance as an evaluated QC (yellow row). This indicates the Relevance QC's role as a potential confounding factor.

## D  Licenses

All system summaries, reference summaries and human-annotated ratings were taken from the SummEval repository under the MIT license. Some of the reference summaries are originally from the CNN/DailyMail dataset. The documents corresponding to the summaries, also from CNN/DailyMail, were retrieved via the Huggingface distribution. All CNN/DailyMail data is released under the Apache-2.0 License.

| Original | *Coherence* **Shuffle Sentences** | *Consistency* **Replace PER** | *Fluency* **Lemmatize Verbs** |
|---|---|---|---|
| Paul Merson *has restarted* his row with andros townsend . The Tottenham midfielder *was brought* on with only seven minutes *remaining* in his team 's 0-0 draw with Burnley. Andros Townsend scores England 's equaliser in their 1-1 friendly draw with Italy in Turin . | The Tottenham midfielder was brought on with only seven minutes remaining in his team 's 0-0 draw with Burnley . Andros Townsend scores England 's equaliser in their 1-1 friendly draw with Italy in Turin . Paul Merson has restarted his row with Andros Townsend . | Roy Hodgson has restarted his row with Andros Townsend . The Merson midfielder was brought on with only seven minutes remaining in his team 's 0-0 draw with Burnley . Andros Townsend scores England 's equaliser in their 1-1 friendly draw with Italy in Turin . | Paul Merson *have restart* his row with Andros Townsend . The Tottenham midfielder *be bring* on with only seven minutes *remain* in his team 's 0-0 draw with Burnley . Andros Townsend scores England 's equaliser in their 1-1 friendly draw with Italy in Turin . |

Table 9: An example of a system summary with all corruption types. A replaced PERSON named entity is marked in red. A lemmatized verb is marked in *blue*. Capital letters were inserted manually to facilitate reading.

| Metric | Coherence | Consistency | Fluency | Relevance |
|---|---|---|---|---|
| ROUGE-1 | 0.9873 | 0.6668 | 0.8966 | N/A |
| ROUGE-2 | 0.9467 | 0.5746 | 0.9121 | N/A |
| ROUGE-3 | 0.9495 | 0.5157 | 0.9128 | N/A |
| ROUGE-4 | 0.9333 | 0.4924 | 0.7202 | N/A |
| ROUGE-L | 0.7566 | 0.7873 | 0.8087 | N/A |
| ROUGE-su* | 0.9766 | 0.7332 | 0.8916 | N/A |
| ROUGE-w | 0.7739 | 0.6902 | 0.8969 | N/A |
| ROUGE-we-1 | 0.9700 | 0.6850 | 0.9958 | N/A |
| ROUGE-we-2 | 0.7927 | 0.7886 | 0.5013 | N/A |
| ROUGE-we-3 | 0.7065 | 0.8060 | 2.1853 | N/A |
| $S^3$-pyr | 0.7438 | 0.6577 | 0.6174 | N/A |
| $S^3$-resp | 0.7828 | 0.6336 | 0.7507 | N/A |
| BertScore-p | 0.8962 | 0.6265 | 0.2925 | N/A |
| BertScore-r | 0.9659 | 0.5216 | 0.4641 | N/A |
| BertScore-f | 0.9786 | 0.5970 | 0.4343 | N/A |
| MoverScore | 0.8043 | 0.5236 | 0.4258 | N/A |
| SMS | 0.6851 | 0.2668 | 0.2962 | N/A |
| BLEU | 0.9598 | 0.3942 | 0.3544 | N/A |
| CHRF | 0.8419 | 0.7789 | 0.7533 | N/A |
| CIDEr | 0.0546 | 15.7989 | 0.1060 | N/A |
| METEOR | 0.8080 | 0.5610 | 0.5993 | N/A |
| SummaQAˆ | 0.9738 | 0.4100 | 0.3489 | N/A |
| BLANCˆ | 0.7650 | 0.3520 | 0.5176 | N/A |
| SuPERTˆ | 0.9157 | 0.2762 | 0.3375 | N/A |
| Novel unigramˆ | 0.1920 | 0.0255 | 0.0368 | N/A |
| Novel bi-gramˆ | 0.5511 | 0.1119 | 0.0742 | N/A |
| Novel tri-gramˆ | 0.3208 | 0.1185 | 0.0991 | N/A |
| Repeated unigramˆ | 0.0114 | 0.3146 | 0.1355 | N/A |
| Repeated bi-gramˆ | 0.1406 | 0.3776 | 0.3073 | N/A |
| Repeated tri-gramˆ | 0.1370 | 0.4934 | 0.1540 | N/A |

Table 10: Absolute relative difference between original performance and bucketing performance anchored by *Relevance*.

| Metric | Coherence | Consistency | Fluency | Relevance |
|---|---|---|---|---|
| ROUGE-1 | N/A | 0.2875 | 0.5543 | 0.1741 |
| ROUGE-2 | N/A | 0.1757 | 0.4923 | 0.1103 |
| ROUGE-3 | N/A | 0.2024 | 0.6400 | 0.2105 |
| ROUGE-4 | N/A | 0.2132 | 0.5158 | 0.2139 |
| ROUGE-L | N/A | 0.2682 | 0.3730 | 0.0694 |
| ROUGE-su* | N/A | 0.3403 | 0.5799 | 0.1774 |
| ROUGE-w | N/A | 0.1865 | 0.3761 | 0.0814 |
| ROUGE-we-1 | N/A | 0.3266 | 0.6588 | 0.1890 |
| ROUGE-we-2 | N/A | 0.3184 | 0.8864 | 0.1472 |
| ROUGE-we-3 | N/A | 0.3165 | 0.0882 | 0.1278 |
| $S^3$-pyr | N/A | 0.1955 | 0.6394 | 0.1139 |
| $S^3$-resp | N/A | 0.1842 | 0.5944 | 0.1109 |
| BertScore-p | N/A | 0.4076 | 0.2373 | 0.1252 |
| BertScore-r | N/A | 0.1969 | 0.3071 | 0.2121 |
| BertScore-f | N/A | 0.3136 | 0.3128 | 0.1613 |
| MoverScore | N/A | 0.3076 | 0.3440 | 0.1792 |
| SMS | N/A | 0.2328 | 0.2847 | 0.3521 |
| BLEU | N/A | 0.1242 | 0.0914 | 0.0771 |
| CHRF | N/A | 0.4792 | 0.7247 | 0.3331 |
| CIDEr | N/A | 17.7140 | 0.1139 | 0.1322 |
| METEOR | N/A | 0.1712 | 0.3405 | 0.1287 |
| SummaQAˆ | N/A | 0.1774 | 0.1952 | 0.1933 |
| BLANCˆ | N/A | 0.1668 | 0.3377 | 0.2045 |
| SuPERTˆ | N/A | 0.1559 | 0.2265 | 0.2771 |
| Novel unigramˆ | N/A | 0.1366 | 0.2128 | 0.9964 |
| Novel bi-gramˆ | N/A | 0.0239 | 0.0029 | 0.2730 |
| Novel tri-gramˆ | N/A | 0.0456 | 0.0386 | 0.5205 |
| Repeated unigramˆ | N/A | 0.9790 | 0.6875 | 1.3634 |
| Repeated bi-gramˆ | N/A | 0.6764 | 0.7280 | 0.6880 |
| Repeated tri-gramˆ | N/A | 0.7937 | 0.8259 | 0.7572 |

Table 11: Absolute relative difference between original performance and bucketing performance anchored by *Coherence*.

| Metric | Coherence | Consistency | Fluency | Relevance |
|---|---|---|---|---|
| ROUGE-1 | 0.1788 | N/A | 0.6285 | 0.1240 |
| ROUGE-2 | 0.2036 | N/A | 0.7033 | 0.1575 |
| ROUGE-3 | 0.2259 | N/A | 0.8096 | 0.1703 |
| ROUGE-4 | 0.2597 | N/A | 0.6754 | 0.1877 |
| ROUGE-L | 0.1708 | N/A | 0.4912 | 0.0983 |
| ROUGE-su* | 0.2094 | N/A | 0.6218 | 0.1400 |
| ROUGE-w | 0.2149 | N/A | 0.5807 | 0.1112 |
| ROUGE-we-1 | 0.1732 | N/A | 0.7033 | 0.1244 |
| ROUGE-we-2 | 0.1617 | N/A | 0.8458 | 0.1100 |
| ROUGE-we-3 | 0.1901 | N/A | 0.3386 | 0.0913 |
| $S^3$-pyr | 0.2117 | N/A | 0.7951 | 0.1051 |
| $S^3$-resp | 0.2179 | N/A | 0.7999 | 0.1154 |
| BertScore-p | 0.2282 | N/A | 0.3091 | 0.2205 |
| BertScore-r | 0.2380 | N/A | 0.3994 | 0.1521 |
| BertScore-f | 0.2373 | N/A | 0.4073 | 0.1574 |
| MoverScore | 0.2043 | N/A | 0.3979 | 0.1413 |
| SMS | 0.2951 | N/A | 0.3782 | 0.2810 |
| BLEU | 0.3225 | N/A | 0.3063 | 0.2167 |
| CHRF | 0.1540 | N/A | 0.4818 | 0.1147 |
| CIDEr | 0.7280 | N/A | 0.4464 | 3.5150 |
| METEOR | 0.2585 | N/A | 0.4722 | 0.1406 |
| SummaQAˆ | 0.5407 | N/A | 0.4021 | 0.3948 |
| BLANCˆ | 0.5268 | N/A | 0.6045 | 0.2827 |
| SuPERTˆ | 0.5649 | N/A | 0.5151 | 0.4122 |
| Novel unigramˆ | 0.7754 | N/A | 0.5059 | 4.2404 |
| Novel bi-gramˆ | 1.3341 | N/A | 0.3823 | 0.3147 |
| Novel tri-gramˆ | 0.7344 | N/A | 0.3851 | 0.7910 |
| Repeated unigramˆ | 0.1479 | N/A | 0.4639 | 0.8627 |
| Repeated bi-gramˆ | 0.1379 | N/A | 0.4344 | 0.3066 |
| Repeated tri-gramˆ | 0.0485 | N/A | 0.5129 | 0.1521 |

Table 12: Absolute relative difference between original performance and bucketing performance anchored by *Consistency*.

| Metric | Coherence | Consistency | Fluency | Relevance |
|---|---|---|---|---|
| ROUGE-1 | 0.1913 | 0.3133 | N/A | 0.0859 |
| ROUGE-2 | 0.2077 | 0.2974 | N/A | 0.0749 |
| ROUGE-3 | 0.1677 | 0.2251 | N/A | 0.0594 |
| ROUGE-4 | 0.2238 | 0.2356 | N/A | 0.0834 |
| ROUGE-L | 0.2303 | 0.3583 | N/A | 0.0729 |
| ROUGE-su* | 0.1984 | 0.3625 | N/A | 0.0919 |
| ROUGE-w | 0.2008 | 0.2693 | N/A | 0.0534 |
| ROUGE-we-1 | 0.1748 | 0.3421 | N/A | 0.0820 |
| ROUGE-we-2 | 0.1192 | 0.2809 | N/A | 0.0349 |
| ROUGE-we-3 | 0.0896 | 0.2595 | N/A | 0.0044 |
| $S^3$-pyr | 0.1251 | 0.1525 | N/A | 0.0205 |
| $S^3$-resp | 0.1483 | 0.1734 | N/A | 0.0286 |
| BertScore-p | 0.4468 | 0.8169 | N/A | 0.2524 |
| BertScore-r | 0.2830 | 0.3041 | N/A | 0.1017 |
| BertScore-f | 0.3259 | 0.5184 | N/A | 0.1306 |
| MoverScore | 0.2813 | 0.4900 | N/A | 0.1438 |
| SMS | 0.3156 | 0.2286 | N/A | 0.1994 |
| BLEU | 0.5043 | 0.3209 | N/A | 0.1893 |
| CHRF | 0.1778 | 0.4029 | N/A | 0.0921 |
| CIDEr | 0.2162 | 212.7716 | N/A | 7.1651 |
| METEOR | 0.2902 | 0.2951 | N/A | 0.0764 |
| SummaQAˆ | 0.5623 | 0.3900 | N/A | 0.3046 |
| BLANCˆ | 0.5358 | 0.3069 | N/A | 0.2018 |
| SuPERTˆ | 0.5096 | 0.2679 | N/A | 0.2778 |
| Novel unigramˆ | 0.7387 | 0.2786 | N/A | 2.8030 |
| Novel bi-gramˆ | 0.8233 | 0.2192 | N/A | 0.9587 |
| Novel tri-gramˆ | 0.9199 | 0.2140 | N/A | 0.7485 |
| Repeated unigramˆ | 0.3130 | 0.7697 | N/A | 0.5830 |
| Repeated bi-gramˆ | 0.1783 | 0.5826 | N/A | 0.3677 |
| Repeated tri-gramˆ | 0.1338 | 0.4467 | N/A | 0.2944 |

Table 13: Absolute relative difference between original performance and bucketing performance anchored by *Fluency*.