# OpenReview forum: "Re-Examining Summarization Evaluation across Multiple Quality Criteria"
_EMNLP/2023/Conference — EMNLP 2023 Findings_

### Official Review · Reviewer_rPrT · 2023-08-02

**Soundness:** 4

**Excitement:**

4: Strong: This paper deepens the understanding of some phenomenon or lowers the barriers to an existing research direction.

**Paper Topic And Main Contributions:**

This paper considers the evaluation metrics across multiple Quality Criteria (QS) in the case of summarization, where QS includes relevance, consistency, fluency, and coherence.
The authors discovered that the conventional setup that uses a high correlation between metric and human scores as an effective metric is inappropriate for multiple QS setting for the summarization case. To achieve this the authors artificially corrupt SummEval system summaries in various forms and show that even the best metrics for each QS fail to detect even drastic summary corruptions with respect to the considered QS. Additionally, they show that there are spurious correlations in a multiple QS setting. The authors also suggest a method of detecting metric-human unreliable correlations scores that are suspected as spurious that removes the effect of the confounding variable.

**Questions For The Authors:**

175. How many times did you spend to corrupt the system summaries?

**Reasons To Accept:**

The paper can be accepted for some reasons: the authors offer a new perspective on the summarization evaluation metrics in the multiple QS case. While the paper mostly highlights the problem, the authors show other researchers to develop additional approaches to evaluate summarization. The title and abstract are correctly selected and written. The references are relevant. The paper is correctly structured and written.

**Reasons To Reject:**

The authors conducted experiments only on SummEval bencmark, it is possible that for other benchmarks, this experiment will give different results. They explain this limitation in the Limitations section, but I didn’t have enough reasoning about bencmarks in other languages and how to get around these limitations for other bencmarks.

**Reproducibility:**

3: Could reproduce the results with some difficulty. The settings of parameters are underspecified or subjectively determined; the training/evaluation data are not widely available.

**Reviewer Confidence:**

3: Pretty sure, but there's a chance I missed something. Although I have a good feel for this area in general, I did not carefully check the paper's details, e.g., the math, experimental design, or novelty.

---

> ### Author Rebuttal · Authors · 2023-08-28
>
> Thank you for your comments!
>
> &nbsp;
>
> **Robustness to other benchmarks**
>
> We demonstrated our methodology over SummEval because it is currently the most prominent dataset which includes human evaluations for multiple Quality Criteria (QCs), applied to the output of a large number of systems. However, our methodology is generically designed to test for spurious correlations across QCs (via our stratified bucketing methodology), and could be applied over any dataset for assessing summarization evaluation metrics, which would include human evaluations for multiple QCs. In case such a dataset, in another language or domain, will have different correlation patterns than SummEval (e.g. weaker correlations for certain QCs), then these are expected to be revealed as well by our bucketing methodology.
>
> &nbsp;
>
> **How many times did we try to corrupt the systems?**
>
> We are not sure whether we correctly interpret your question. As far as we understand, the question refers to the effort needed to identify corruptions that are missed by the considered evaluation metrics. In fact, this was rather a small effort, where it was easy to design substantial corruptions that were overlooked by the evaluation metrics. These corruptions were applied for all summaries for each system in turn.

---

### Official Review · Reviewer_jPgs · 2023-08-04

**Soundness:** 2

**Excitement:**

2: Mediocre: This paper makes marginal contributions (vs non-contemporaneous work), so I would rather not see it in the conference.

**Paper Topic And Main Contributions:**

This paper discusses the reliability of automatic evaluation metric. The experimental results suggests that scores obtained from an automatic metric designed to evaluate a specific Quality Criteria (QC) are not reliable in the other QCs. This paper recommends that an automatic metric designed for assessing a specific QC should only be used for evaluating text in relation to that particular QC.

**Reasons To Accept:**

The findings of this paper are helpful for interpretation of automatic evaluations.

**Reasons To Reject:**

The claim of this paper, an automatic evaluation metric designed to evaluate text in terms of a specific QC should not be utilized to assess text in terms of the other QCs, is helpful but not surprising. According to the nature of ROUGE-N, many researchers do not utilize it to evaluate the coherency of the summaries.

Finding confounding factors through experiments can be a complex and confusing process. In my opinion, utilizing partial correlations can prove to be beneficial for this purpose.

**Reproducibility:**

4: Could mostly reproduce the results, but there may be some variation because of sample variance or minor variations in their interpretation of the protocol or method.

**Reviewer Confidence:**

4: Quite sure. I tried to check the important points carefully. It's unlikely, though conceivable, that I missed something that should affect my ratings.

---

> ### Author Rebuttal · Authors · 2023-08-28
>
> Thank you for your comments!
>
> &nbsp;
>
> **Novelty and importance of our approach**
>
> As indicated in the review, we indeed conclude in our paper that each evaluation metric should focus on a single QC. While this insight is not surprising to the reviewer, a prominent thread of research in NLP has been effectively advocating the opposite. As we point out, the SummEval paper, which is highly cited (~330 citations as of today) and is one of the most prominent current datasets for evaluating summarization evaluation metrics, clearly implies that metrics *can* be suitable for evaluating multiple QCs, and establish such metric fitness only by computing a direct correlation independently with each QC. Consequently, for example, they explicitly specify Rouge-N (for large N) to be better than Rouge-L in all QCs, while we claim that most of these scores are spurious. Furthermore, many papers suggesting new metrics do follow the SummEval methodology and data and show high correlation across all QCs to their new metric (e.g.,DiscoScore [1], BARTScore [2], InfoLM [3]) - again counter to our findings. Some other papers point to the advantage of a specific metric for a specific QC according to the SummEval results. For example, in [4]  they explained their results by the ability of Rouge-3 to measure consistency as resulted in SummEval, while in [5] they rely on SummEval to conclude that “ROUGE-L can be seen as a measure for fluency”.
>
> Thus, even if our findings are not surprising to some people in the community, they are meant to respond to this notable literature thread, offering three contributions for future research: (1) exposing the fallacies of previous conclusions, which inappropriately attributed metrics to certain QCs; (2) revealing underlying correlations in manual summarization evaluation data, and showing accordingly that prior false conclusions stemmed from spurious correlations caused by confounding factors; (3) offering a concrete experimental protocol, based on a data stratification approach (bucketing), which helps detecting spurious correlations (elaborated next in response to the second reason to reject). We believe that these contributions, in the context of the negative effects of highly visible practices in the literature, would be beneficial to the community and are hence important to publish.
>
> &nbsp;
>
> **Bucketing versus partial correlation**
>
> When examining the statistics literature for methods for detecting spurious correlations, we indeed focused on the two prominent approaches of computing partial correlation vs. data stratification (followed in our bucketing method), and experimented with both.
>
> Methodologically, we identified two advantages of bucketing. First, beyond providing an overall “debiased” correlation figure, this method provides explicitly the debaised data within each bucket, allowing for further interpretability and analysis (e.g. examining correlation and other statistics within each bucket). Second, the partial correlation calculation is based on computing linear regression between the variables, assuming that the potential correlation between them is linear. Bucketing, on the other hand, generally allows for measuring any type of correlation, making this method more general for future extensions (even if in the current work we measured only linear correlations within each bucket).
>
> Empirically, as said, we did compute partial correlations in our investigations, but found them to be less discriminative in detecting spurious correlations in the summarization data, although showing the same trend. Following your comment, we will add to the paper a presentation and empirical results of the partial correlation method, and compare it to the bucketing method.
>
> &nbsp;
>
> [1] Zhao, Wei, Michael Strube and Steffen Eger. “DiscoScore: Evaluating Text Generation with BERT and Discourse Coherence.” Conference of the European Chapter of the Association for Computational Linguistics (2022).
>
> [2] Yuan, Weizhe, Graham Neubig, and Pengfei Liu. "BARTScore: Evaluating generated text as text generation." Advances in Neural Information Processing Systems 34 (2021): 27263-27277.‏
>
> [3] Colombo, Pierre Jean A., Chloé Clavel, and Pablo Piantanida. "InfoLM: A new metric to evaluate summarization & data2text generation." Proceedings of the AAAI Conference on Artificial Intelligence. Vol. 36. No. 10. 2022.‏
>
> [4] Saadia Gabriel, Asli Celikyilmaz, Rahul Jha, Yejin Choi, and Jianfeng Gao. ``GO FIGURE: A Meta Evaluation of Factuality in Summarization.” In Findings of the Association for Computational Linguistics: ACL-IJCNLP 2021, pages 478–487.
>
> [5] Li, Daniel, et al. "Improving Automatic Summarization for Browsing Longform Spoken Dialog." Proceedings of the 2023 CHI Conference on Human Factors in Computing Systems. 2023.‏

---

### Official Review · Reviewer_bKiH · 2023-08-05

**Soundness:** 4

**Excitement:**

3: Ambivalent: It has merits (e.g., it reports state-of-the-art results, the idea is nice), but there are key weaknesses (e.g., it describes incremental work), and it can significantly benefit from another round of revision. However, I won't object to accepting it if my co-reviewers champion it.

**Paper Topic And Main Contributions:**

The authors study the framework of summarization evaluation along four quality criteria (QC; consistency, faithfulness,  relevance and coherence) and challenge the current method of studying correlations between human judgements and metric-based ratings as a measure of an automatic metric quality. The authors find that some of the best metrics for certain QC actually do not perform well, failing to detect even drastic summary corruptions with respect to the considered QC. For instance, changing the word order does not affect a metric's raking on coherence criteria. To explain this finding, they study the correlations between these four criteria and find that human annotations are correlated because one of the criteria act as a confounder. Thus, reporting correlations with human judgements after annulling for this confounder can help better estimate system performance instead of studying raw correlations. Authors propose a bucketing system where they divide the ratings into buckets such that the confounder variable has same rating, lending an effect of keeping the confounder constant. The authors also find that most often, relevance QC acts as a confounder since it was mainly used to design systems/metrics.

**Questions For The Authors:**

My following question is intended as a discussion with the authors.
1. While reading this paper, it motivated me to think conceptually about the four metrics and their correlations just based on their definitions. For instance, theoretically a highly coherent summary (quality of the collective structure of the sentences in the summary) can still have low fluency (linguistic quality of individual sentences in the summary) if the individual words are shuffled in the sentence while preserving the overall discourse structure. However, the author's findings suggest that, in the dataset they analyzed, they observe a high correlation between these dimensions. Could this be explained by the fact that it is very less likely to observe such cases in practice, i.e., in this dataset, if the summary has a good discourse structure (high coherence), it is likely it will also have high fluency (words in the sentences are ordered such that they make sense)? Does this observation mean that all the model-generated summaries in the benchmark have summaries such that, if they are coherent, they are also fluent? I think it is possible to draw theoretical insights based on the definition of the quality criteria and contrast them with empirical findings, which could reveal some interesting findings.

**Reasons To Accept:**

The paper raises an important concern in automatic evaluation of summarization, i.e., use of multiple criteria for evaluation and their correlations. The authors demonstrate the brittleness of current evaluation practices in the presence of summary corruptions and propose a procedure to help detect such issues. Overall the paper is well written and provides experiments to support their findings related to issues with current evaluation practices.

Suggestion:
1.  While I understand the page limit constraints, if possible, the paper could benefit from showing some worked out examples to ease reading and comprehension of the paper.
2. Also, some examples of how the ratings changed when the summaries were corrupted could be interesting.

**Reasons To Reject:**

I don't think I have any major concerns.

**Reproducibility:**

3: Could reproduce the results with some difficulty. The settings of parameters are underspecified or subjectively determined; the training/evaluation data are not widely available.

**Reviewer Confidence:**

3: Pretty sure, but there's a chance I missed something. Although I have a good feel for this area in general, I did not carefully check the paper's details, e.g., the math, experimental design, or novelty.

---

> ### Author Rebuttal · Authors · 2023-08-28
>
> Thank you for your comments!
>
> &nbsp;
>
> **Suggestions for adding illustrative example**
>
> Thank you for these good suggestions, we will add such examples.
>
> &nbsp;
>
> **Discussion about inherent correlation between QCs**
>
> This is a very interesting question. Within the scope of the paper, we indeed observed a high correlation across QCs in the SummEval data, and attributed it to the assumption that more powerful systems tend to perform better under all criteria. WIth respect to a theoretical perspective on inherent correlations between the QCs, on the face of it it seems that the designers of these QCs did have in mind a substantial degree of independence between them. For example, any extractive summarization system would inherently be quite fluent, as it consists of original source sentences, each manually written (up to decontextualization issues). Yet, one can imagine an extractive system, particularly in the multi-document setting, which is quite poor wrt the generated discourse structure. Similarly, an extractive system would tend to be consistent (faithful), but may perform poorly wrt detecting salient content (i.e. poor relevance). That said, some dependencies may still exist inherently between QCs, per their definitions. For example, for a sentence to be fluent, discourse references and connectives should be easily interpretable, which would rely on a coherent discourse structure, also across sentences. Similarly, a non-consistent (unfaithful) text fragment, which is not entailed by the source, cannot reflect relevant content in that source (as its content is not at all present in the source). Investigating such varying degrees of QC dependencies indeed seems like an interesting topic for future research.

---

### Meta-Review · Area_Chair_9FwE · 2023-09-19

**Recommendation:** 4

**Metareview:**

The paper shows that high correlations between system rankings from metrics and humans can be spurious, in the presence of one confounding QC. The arguments in the paper are convincing. The authors should consider addition of results from partial correlation, which they claim shows similar findings to strengthen the paper.

---

### Decision · Program_Chairs · 2023-10-07

**Decision:**

Accept-Findings

**Comment:**

The paper shows that high correlations between system rankings from metrics and humans can be spurious, in the presence of one confounding QC. The arguments in the paper are convincing. The authors should consider addition of results from partial correlation, which they claim shows similar findings to strengthen the paper.